# A cautionary note on the use of unsupervised machine learning algorithms to characterise malaria parasite population structure from genetic distance matrices

**James A. Watson** [1,2]*, **Aimee R. Taylor** [3,4], **Elizabeth A. Ashley** [2,5], **Arjen Dondorp** [1,2], **Caroline O. Buckee** [3], **Nicholas J. White** [1,2], **Chris C. Holmes** [6,7]

**1** Mahidol-Oxford Tropical Medicine Research Unit, Faculty of Tropical Medicine, Mahidol University, Bangkok, Thailand, **2** Centre for Tropical Medicine and Global Health, Nuffield Department of Medicine, University of Oxford, Oxford, United Kingdom, **3** Center for Communicable Disease Dynamics, Department of Epidemiology, Harvard T. H. Chan School of Public Health, Boston, Massachusetts, USA, **4** Broad Institute of MIT and Harvard, Cambridge, Massachusetts, USA, **5** Lao-Oxford-Mahosot Hospital Wellcome Trust Research Unit, Vientiane, Laos, **6** Department of Statistics, University of Oxford, Oxford, United Kingdom, **7** Nuffield Department of Medicine, University of Oxford, Oxford, United Kingdom

\* jwatowatson@gmail.com

**Data Availability Statement:** All intermediate data are available at https://doi.org/10.5281/zenodo.4023373. Raw genetic data are available from the

## Abstract

Genetic surveillance of malaria parasites supports malaria control programmes, treatment guidelines and elimination strategies. Surveillance studies often pose questions about malaria parasite ancestry (e.g. how antimalarial resistance has spread) and employ statistical methods that characterise parasite population structure. Many of the methods used to characterise structure are unsupervised machine learning algorithms which depend on a genetic distance matrix, notably principal coordinates analysis (PCoA) and hierarchical agglomerative clustering (HAC). PCoA and HAC are sensitive to both the definition of genetic distance and algorithmic specification. Importantly, neither algorithm infers malaria parasite ancestry. As such, PCoA and HAC can inform (e.g. via exploratory data visualisation and hypothesis generation), but not answer comprehensively, key questions about malaria parasite ancestry. We illustrate the sensitivity of PCoA and HAC using 393 *Plasmodium falciparum* whole genome sequences collected from Cambodia and neighbouring regions (where antimalarial resistance has emerged and spread recently) and we provide tentative guidance for the use and interpretation of PCoA and HAC in malaria parasite genetic epidemiology. This guidance includes a call for fully transparent and reproducible analysis pipelines that feature (i) a clearly outlined scientific question; (ii) a clear justification of analytical methods used to answer the scientific question along with discussion of any inferential limitations; (iii) publicly available genetic distance matrices when downstream analyses depend on them; and (iv) sensitivity analyses. To bridge the inferential disconnect between the output of non-inferential unsupervised learning algorithms and the scientific questions of interest, tailor-made statistical models are needed to infer malaria parasite ancestry. In the absence of such models speculative reasoning should feature only as discussion but not as results.

Pf3K release: (https://www.malariagen.net/projects/pf3k).

**Funding:** This study was part of the Mahidol University Oxford Tropical Medicine Research Programme funded by the Wellcome Trust of the United Kingdom (core grant 106698/B/14/Z). The clinical studies were supported by the United Kingdom Department for International Development (programme code 201900). Genotyping for this study was funded by the Wellcome Trust (098051, 206194), Bill \& Melinda Gates Foundation (OPP11188166) and the Medical Research Council (G0600718). Genome sequencing was done by the Wellcome Sanger Institute (WSI). A.R.T. and C.O.B. are supported by a NIGMS Maximizing Investigators' Research Award (MIRA) for Early Stage Investigators (R35GM-124715-02). The funders had no role in study design, data collection and analysis, decision to publish, or preparation of the manuscript.

**Competing interests:** The authors have declared that no competing interests exist.

## Author summary

Genetic epidemiology studies of malaria attempt to characterise what is happening in malaria parasite populations. In particular, they are an important tool to track the spread of drug resistance and to validate genetic markers of drug resistance. To make sense of parasite genetic data, researchers usually characterise the population structure using statistical methods. This is most often done as a two step process. The first is a data reduction step, whereby the data are summarised into a distance matrix (each entry represents the genetic distance between two isolates). The distance matrix is then input into an unsupervised machine learning algorithm. Principal coordinates analysis and hierarchical agglomerative clustering are the two most popular unsupervised machine learning algorithms used for this purpose in malaria genetic epidemiology. We highlight that this procedure is sensitive to the choice of genetic distance and to the specification of the algorithms. These unsupervised methods are useful for exploratory data analysis but cannot be used to infer historical events. We provide some guidance on how to make genetic epidemiology analyses more transparent and reproducible.

## Introduction

As part of a global push to eliminate malaria, there are ongoing efforts to roll out routine molecular surveillance to understand how the parasites causing malaria are spreading, to track drug resistance and identify its emergence, and to understand where to target interventions [1]. Malaria parasite genetic data have been accrued increasingly rapidly in recent years, but the analytical methods for making sense of them lag behind. Many of the key questions posed by studies of malaria parasite genetic epidemiology can be phrased in terms of population structure and thus parasite ancestry. Seminal methods for characterising and quantifying population structure in general include algorithms such as EIGENSTRAT, STRUCTURE, fineSTRUCTURE and ADMIXTURE. EIGENSTRAT provides a continuous estimate of population structure (a projection onto a linear combination of genetic variants) and is typically used in the context of genome wide association studies to adjust for population stratification [2]. Methods such as STRUCTURE [3], fineSTRUCTURE [4] and ADMIXTURE [5] perform population assignment (i.e. assign individuals to one or more inferred ancestral populations). These methods have been developed primarily in the context of human statistical genetics. Their empirical success in human genetics does not directly imply that they are applicable to population genetic analyses of malaria parasites: eukaryotic pathogens of the genus *Plasmodium* that are transmitted by anopheline mosquitoes. Malaria parasites are haploid during the human host stage of their life-cycle, but undergo an obligate stage of sexual recombination in the mosquito. Although both humans and malaria parasites recombine sexually, the added complexity of the malaria parasite life cycle complicates and confounds *Plasmodium* population genetics. For example, unlike humans, malaria parasites can self (recombination between genetically identical male and female gametes) [6], and the rate of selfing varies with transmission intensity [7]. There is a need to test the sensitivity of specific methods in the context of malaria parasite genetics before making conclusive inferences.

Malaria parasite population genetics has borrowed methods from human genetics, but has also heavily relied on context agnostic unsupervised learning algorithms primarily developed in machine learning (e.g. [8–10]). Unsupervised learning algorithms can detect patterns and structure in data without needing classifications or categories as inputs. Principal coordinates

analysis (PCoA) and hierarchical agglomerative clustering (HAC) are the two most popular classes of unsupervised learning algorithms used in genetic epidemiology studies of malaria. PCoA is a generalisation of principal components analysis (PCA). PCA has deep connections with statistical model-based clustering analysis (e.g. STRUCTURE and related methods) [4] and, under certain assumptions, has a genealogical interpretation [11]. In contrast, unsupervised clustering algorithms such as HAC do not, in general, have a genealogical interpretation and are non-inferential (they do not have parameters whose values are inferred from data). This limits the scope of the scientific questions they can answer. They are however important stopgap methods and remain useful conceptual aids.

Both PCoA and HAC operate on a distance matrix. The construction of a genetic distance matrix is a common feature of many computational analysis pipelines used in both human and *Plasmodium* population genetics. Typically, the genetic data for all (or a subset of) polymorphic loci for a set of *n* isolates are summarised into an *n*-by-*n* matrix of scalar distances or similarities (distance is inversely related to similarity). The distance or similarity used is context specific and can either be computed directly as a function of the genetic data or inferred under a statistical model fit to the data [12]. The use of genetic distance or similarity matrices goes beyond PCoA and HAC. For example, fineSTRUCTURE performs statistical model-based clustering of the co-ancestry matrix (a similarity matrix) generated by the chromosome painting algorithm (ChromoPainter) [4]. The assumption in fineSTRUCTURE is that the co-ancestry matrix is a (nearly) sufficient statistic: i.e. all of the information in the data that is needed to infer the parameters of the clustering model is contained in the co-ancestry matrix. In a similar way, PCoA and HAC only use as input the summarised information in the distance matrix. Summarising the data in this succinct way has many advantages. For example, when dealing with whole genome sequence (WGS) data, the number of isolates is usually many orders of magnitude smaller than the number of loci. Thus the distance matrix is computationally easier to analyse (proportional to the square of the number of isolates). In addition it is not possible to reverse engineer the distance matrix to obtain the original genetic data, making it a privacy preserving summary of the genetic data. Careful consideration of which genetic distances should be used to construct this matrix is needed. Making the distance matrix fully available along with the code renders all analyses dependent on this data summary fully reproducible.

## Computational analysis pipelines in malaria genetic epidemiology studies

The computational analysis pipelines of many genetic epidemiology studies of *Plasmodium* spp. roughly follow the steps shown in the schematic outline in Fig 1. Genetic data, for example short read counts aligned to a reference genome are first pre-processed to give a (phased) variant matrix. Pre-processing of the raw data will typically involve many important analytic decisions (how to impute missing data; choice of thresholds for low read counts or low-quality isolates; what to do with polyclonal samples). The output variant matrix (each row is a phased haplotype, each column is a genetic marker) is then summarised as a distance (similarity) matrix, either using a statistical model or from a direct computation on the variant matrix. The distance matrix is then input into methods which can help determine and visualise structure such as PCoA, or can help cluster haplotypes into groups such as fineSTRUCTURE or HAC algorithms. Under this schema, the genetic distance matrix is used as a (nearly) sufficient statistic for all downstream analyses. Box 1 gives an example of population genetic data that have been analysed under this type of pipeline.

This work addresses the sensitivity of final outputs with respect to the choice of genetic distance and to the specification of algorithms that take as input a distance matrix. We do not address the important question of sensitivity to pre-processing choices (this includes allele

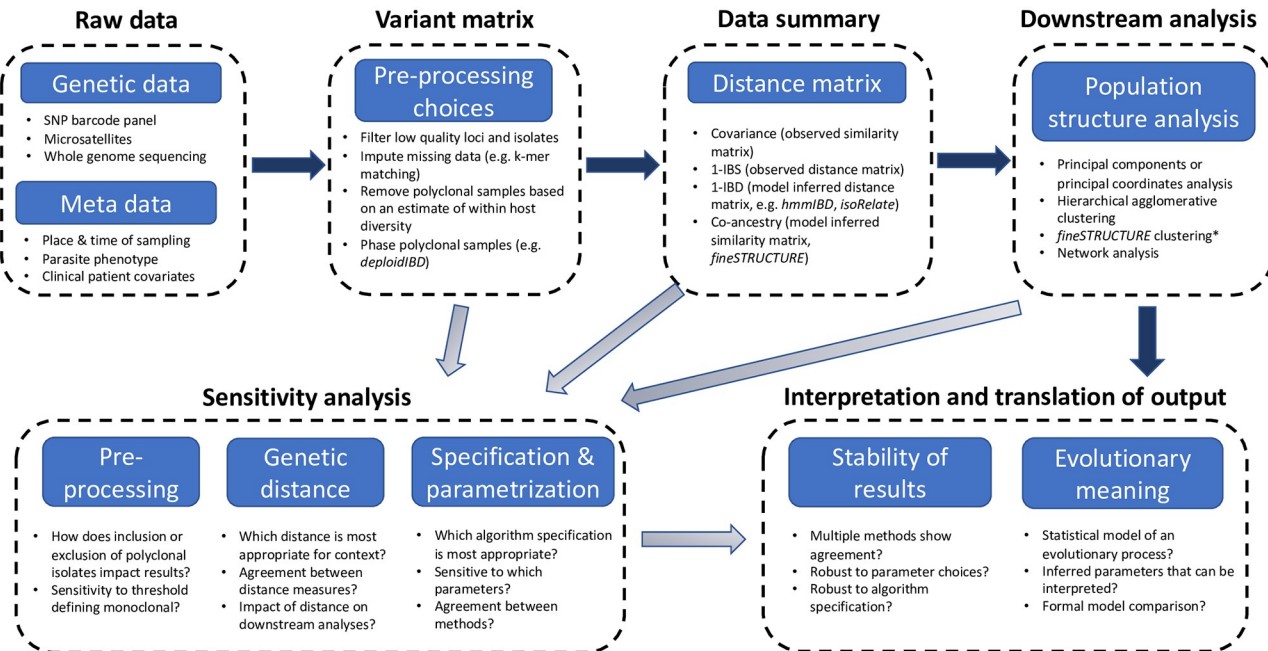

**Fig 1. A rough schematic of a typical computational analysis pipeline in a malaria genetic epidemiology study, whereby the generation of a genetic distance or similarity matrix is a key step in the data analysis.** The main analysis pipeline is shown with the dark arrows. Additional sensitivity analyses feed into the final interpretation and translation of results and are shown by the light arrows. Bullet points show examples of data modalities, data processing algorithms or analytical approaches. *fineSTRUCTURE is itself a computational pipeline that takes as input phased haplotypes (e.g. a phased variant matrix), computes a co-ancestry matrix and then performs clustering on this matrix. fineSTRUCTURE clustering corresponds to the second stage of the fineSTRUCTURE pipeline.

frequency thresholding, SNP detection algorithms, and imputation of missing values). An important problem in most population genetic epidemiology studies of malaria is what to do with complex infections (distinct parasite genomes present in the same human host). There exists software for phasing complex infections which are usually composed of related parasites, e.g. DEploidIBD [7], but this is not often done in practice and the implications of ignoring complex infections are not fully understood ([13] uses simulation to quantify the impact of excluding polyclonal isolates). We undertake an empirical sensitivity analysis of PCoA and HAC with respect to the genetic distance measure used and to the algorithm specification. Our analysis uses sequence data from *P. falciparum* parasites isolated from a subset of patients enrolled in the Eastern GMS sites (Cambodia and neighbouring areas of Thailand, Laos and Vietnam) of a large clinical trial that took place between 2011 and 2013 [14] (see Box 1 for some background providing the rationale for the use of these data). We provide some tentative guidance as to the use and interpretation of PCoA and HAC in the context of *Plasmodium* population genetics, including ways to increase the transparency and reproducibility of analytic pipelines (e.g. by sharing key intermediate data summaries such as genetic distance matrices). In the discussion we make a call for the development of more targeted statistical models for malaria parasite ancestry.

## Results

### Genetic distance between *P. falciparum* monoclonal isolates from Southeast Asia

We computed three distance matrices for 393 whole genome sequences from parasites collected in the Eastern GMS, using as genetic distance $1-$IBS, $1-$IBD, and $-\log_2$IBD, respectively

### Box 1: Motivating example: The spread of multi-drug resistant *P. falciparum* in the Greater Mekong subregion and the use of unsupervised learning algorithms.

Unsupervised learning algorithms applied to genetic distance matrices have been used extensively to characterise and quantify the current epidemic of multi-drug resistant *P. falciparum* in the Greater Mekong subregion (GMS) [8–10, 15]. In the attempt to understand how drug resistance has considerably worsened in the last decade, two observations and the corresponding analytical approaches had considerable importance in the scientific debate.

The first observation, made before the discovery of the causative mutations in the *Pfkelch13* gene which confer artemisinin resistance, was the identification of multiple apparently sympatric sub-populations in the Eastern GMS [8]. These sub-populations were identified primarily by PCoA of a genetic distance matrix based on identity-by-state (IBS; see Methods for definition) [8]. The second observation was the spread of a single multi-drug resistant parasite lineage across the eastern GMS. This observation was first made using a small panel of microsatellite markers flanking the *Pfkelch13* gene which showed that a single long haplotype (bearing the *Pfkelch13* mutation C580Y and named *Pf*Pailin, as it was observed first in the Pailin area of Western Cambodia) had spread across Cambodia into North East Thailand, Southern Laos and Southern Vietnam [16, 17]. This parasite lineage also acquired resistance to piperaquine, manifest by amplification in the *Pfplasmepsin* gene and mutations in *Pfcrt*. A subsequent study used WGS data and presented qualitatively similar results; *Pf*Pailin was renamed the KEL1/PLA1 lineage [9]. The KEL1/PLA1 lineage was based on HAC of a genetic distance matrix derived from chromosome painting [9].

These two observations and associated studies [8–10, 15–17] are of considerable importance as it is in this same region of Southeast Asia that antimalarial drug resistance arose previously, first to chloroquine and then to sulphadoxine-pyrimethamine, before spreading to India and Africa at a cost of millions of lives. *P. falciparum* has now evolved resistance to both artemisinin [14] and to piperaquine [18], the two active components of the artemisinin combination therapy (ACT) dihydroartemisinin-piperaquine [19]. Dihydroartemisinin-piperaquine was the first line treatment for falciparum malaria in Eastern Thailand, Cambodia and Vietnam until worsening resistance forced a change in policy. There has been controversy over the role of artemisinin resistance in facilitating the emergence of partner drug resistance. Understanding whether artemisinin resistance promoted the selection of piperaquine resistance in the Eastern GMS (or mefloquine resistance on the Thailand-Myanmar border), and if so by how much, is of contemporary relevance for antimalarial drug policies and practices. Identification of spatio-temporal patterns and subpopulations via the analysis of genetic data using unsupervised machine learning methods can help answer these questions. However, we believe that there has been a disconnect between the scientific questions of interest and those that methods such as PCoA and HAC can address.

(see Methods for exact definitions). Fig 2 shows the distribution of the distances between pairs of isolates for the three choices of distance measure, and the agreement between the 1-IBS and 1−IBD distances. For closely related pairs (IBD close to 1), 1−IBS and 1−IBD strongly agree. However, when comparing unrelated pairs of parasites (e.g. 1−IBD greater than 0.95), there is

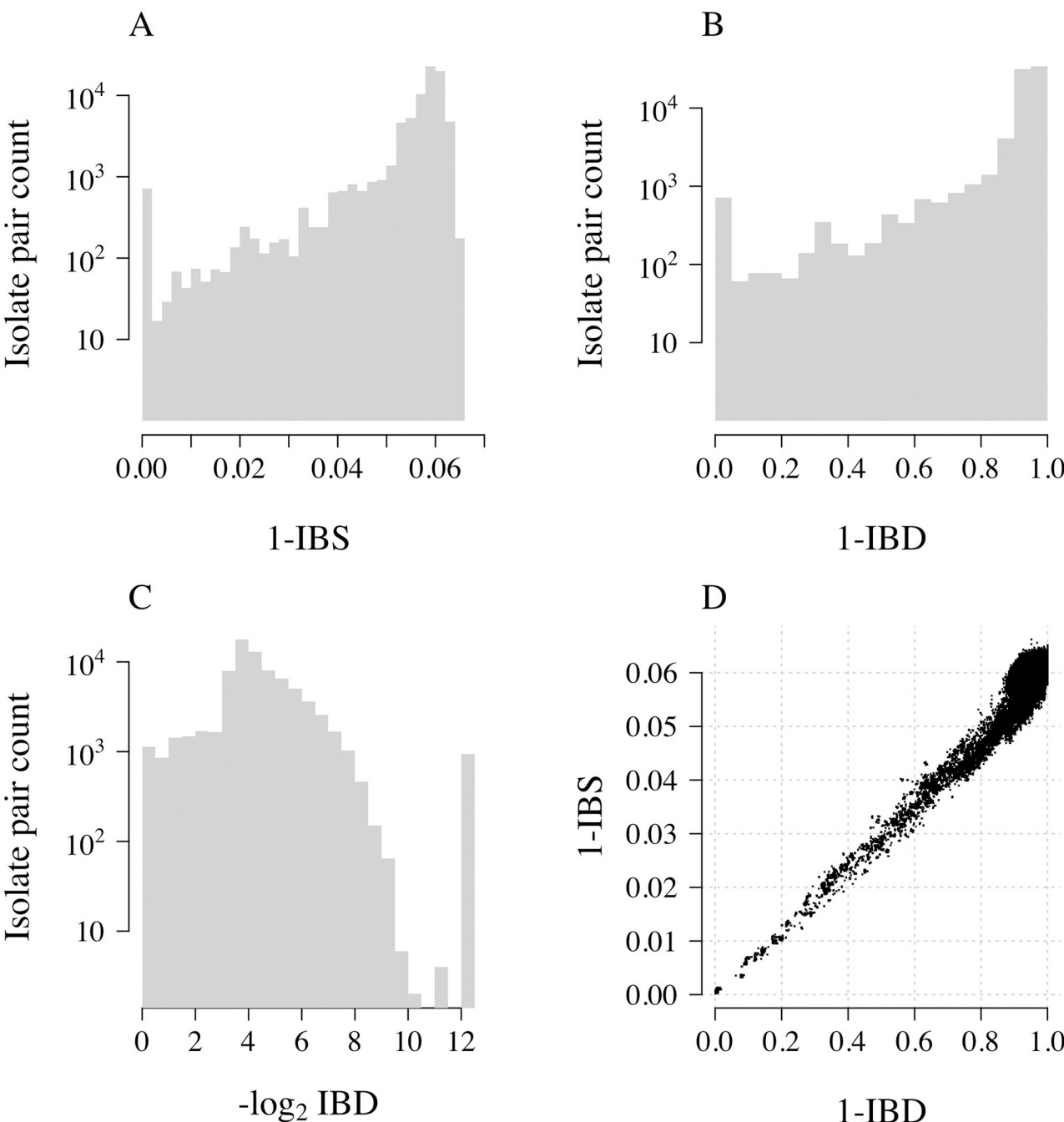

**Fig 2. Comparison of three genetic distances.** Each panel represents distances between 77028 pairs of 393 *P. falciparum* isolates from the Eastern GMS. A: 1−IBS distance; B: 1−IBD distance; C: −log$_2$ IBD distance; D: agreement between 1−IBS and 1−IBD. Note that the y-axes in panels A-C are on a log$_{10}$ scale. The set of distances such that −log$_2$ IBD is approximately 12.5 are those with estimated IBD equal to zero, replaced by a lower limit of quantification equal to the smallest IBD value greater than zero. Note that 1−IBD spans the zero to one range, whereas 1−IBS does not.

considerable variation in the 1−IBS distance (variation across approximately 20% of its observed range).

## Sensitivity of PCoA to the genetic distance

PCoA applied to the three distance matrices gave qualitatively similar results for 1−IBS and 1−IBD, but differed considerably for −$\log_2$ IBD (Fig 3). For both 1−IBS and 1−IBD, the first principal component distinguishes clearly between *Pfkelch13* wild type parasites (green) and *Pfkelch13* mutated parasites (not green and predominantly the artemisinin resistant C580Y mutation shown in blue). This first principal component also distinguishes between *Pfplasmepsin* single copy (circles) and *Pfplasmepsin* amplified (triangles) parasites (associated with piperaquine resistance). This pattern also emerges from the first principal component of the PCoA of the −$\log_2$ IBD distance matrix, while the second shows the same level of variation between the wild type parasites (green circles) as amongst multi-drug resistant parasites with the C580Y mutation (blue triangles). It may be tempting to give an evolutionary interpretation to the PCoA pattern shown in panel C. For example, this PCoA pattern could demonstrate how what was initially a soft sweep became a hard sweep over time (leftmost contains almost exclusively WT parasites, centre contains the many resistant alleles seen initially in the Eastern GMS, rightmost contains almost exclusively C580Y isolates with amplification in *Pfplasmepsin*). This was the interpretation applied earlier to the *Pfkelch13* flanking sequence haplotype data [16]. However, there is no available mathematical theory supporting possible interpretations of the patterns observed in panels A-C of Fig 3. As an aside, there would be for 1-IBS between isolates from disconnected populations that do not have the opportunity to recombine (e.g. on different continents) or 1-IBS based on mitochondrial DNA (since mitochondrial DNA does not recombine) and thus the coalescent model applies [11]. As a sensitivity analysis, we computed the co-ancestry matrix output from the ChromoPainter algorithm (in the fineSTRUCTURE computational pipeline). Panel D of Fig 3 shows PCoA of the co-ancestry matrix which is qualitatively very similar to PCoA on 1−IBS and 1−IBD distance matrices.

PCoA applied to arbitrary genetic distance matrices is a powerful visual tool, but the interpretation of the output is sensitive to the choice of genetic distance.

## Sensitivity of HAC-based discrete clustering

Dendrograms constructed using HAC algorithms do not, in general, approximate the unknown pedigree or phylogeny of the isolates. Therefore the topology of the dendrogram is not of primary interest. In addition, the clustering arrangement which underlies all possible topologies is, in general, highly dependent on the genetic distance and on the HAC algorithm specification. Comparing dendrogram topologies constructed using different genetic distances or algorithm specifications for a large set of parasite isolates is difficult. For example, a common approach is to use tanglegrams (also known as co-phylo plots) which show a side-by-side comparison of leaf placements with joining lines for two dendrograms [20]. The implicit assumption is that the level of entanglement of the lines is proportional to the level of incongruence between the dendrograms. However, the correlation between entanglement and dendrogram incongruence is weak and so tanglegrams can be misleading [21]. A key reason for this is that the ordering of the leaves within the dendrogram is arbitrary. Many leaf orderings correspond to the same clustering arrangement, as shown in Fig 4. The visual topology of the tree is therefore unstable. This is especially important as meta-data are often used to check the coherency of a clustering structure found by HAC, for example by colouring the leaves or by adding a coloured bar at the bottom of the tree. Fig 4 shows the same dendrogram under two possible leaf orderings (given by random permutations). The coloured bars underneath the

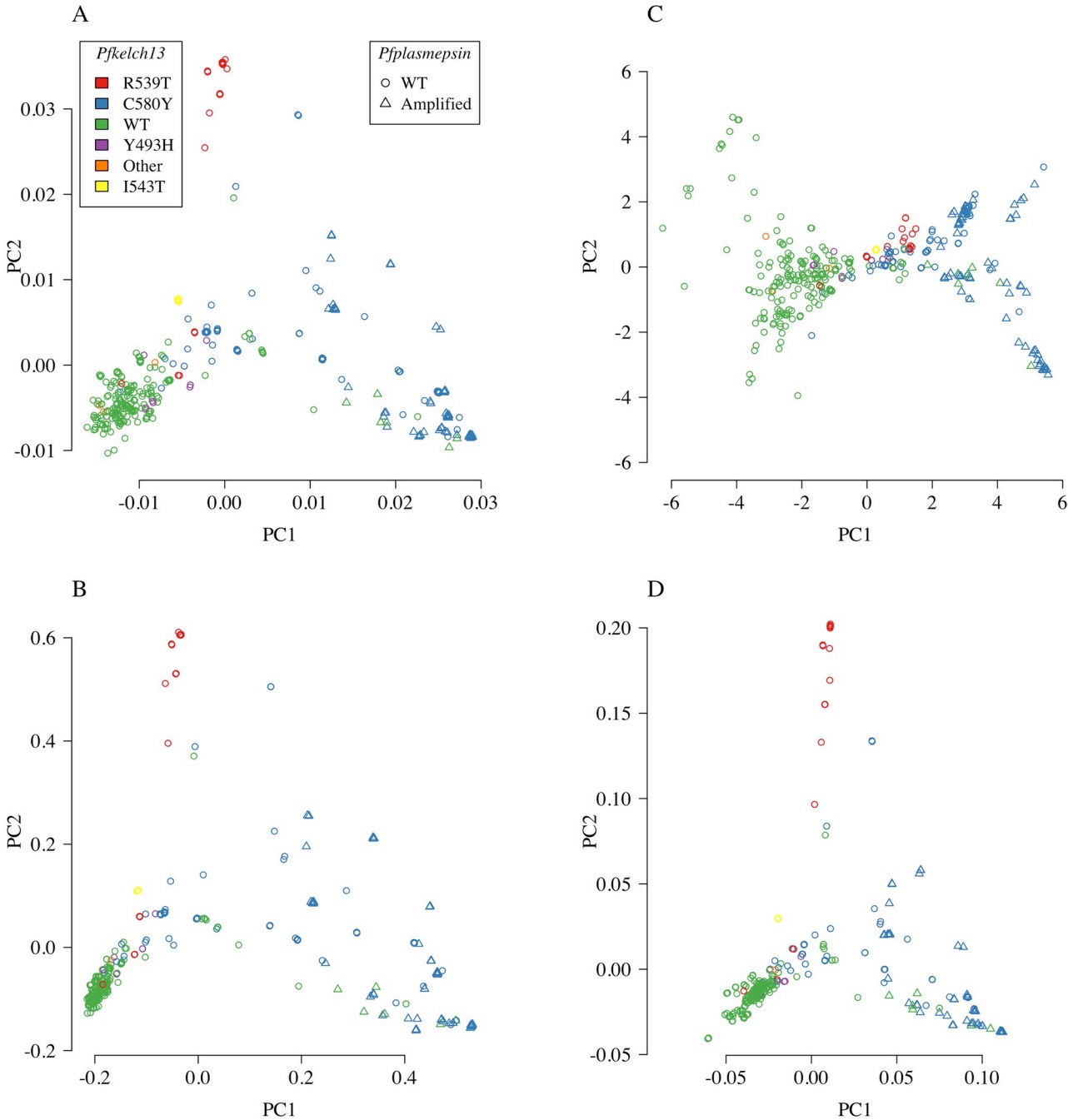

**Fig 3. PCoA and PCA of summary *n*-by-*n* distance and similarity matrices for *n* = 393 isolates.** Panels A-C show PCoA applied to the 1−IBS (A), 1−IBD (B), and −log$_2$ IBD (C) distance matrices. Panel D shows PCA applied to the co-ancestry matrix computed using fineSTRUCTURE version 4. Isolates are plotted along the first two principal components. Colours correspond to the different known causative mutations in the *Pfkelch13* gene, where green is wild type (WT) and blue is C580Y. Triangles correspond to *Pfplasmepsin* amplified parasites, and circles correspond to parasites that are WT in *Pfplasmepsin*.

trees show the *Pfkelch13* mutations and highlight how different the alternative orderings can be. For this reason, it is important to note this sensitivity when superimposing meta-data onto a tree structure. This issue can be avoided with the use of a formal statistical test between the tree structure and meta-data as proposed in [22].

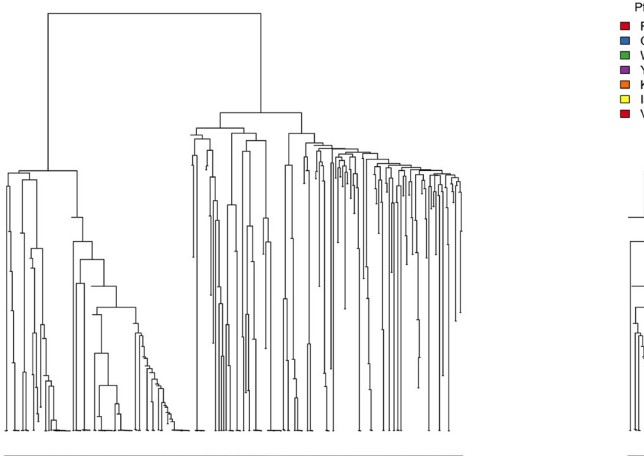
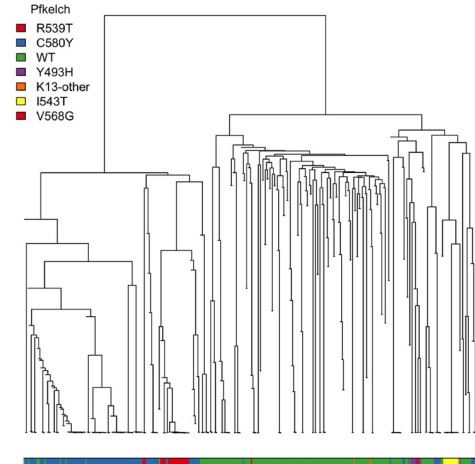

**Fig 4. Two distinct dendrograms which depict the same underlying clustering arrangement.** The ordering of the leaves was changed by randomly rotating the internal nodes. The clustering arrangement was produced by applying HAC with average linkage to the -log₂ IBD distance matrix. The coloured bars below the dendrograms visualise the corresponding *Pfkelch13* mutation of each isolate (green: wild type; blue: C580Y). This illustrates how the ordering of the meta-data is sensitive to arbitrary choices for the dendrogram topology.

Our main focus now, is the sensitivity of discrete cluster assignments based on HAC. A dendrogram (which is a visualisation of a set of nested clustering arrangements) can be collapsed into discrete clusters by choosing a cut-point on the y-axis (a specific branch height). For example, the KEL1 lineage reported in [9] was determined on the basis of cutting an HAC dendrogram into a set of discrete clusters. The largest cluster (named KEL1) was shown to be dispersed across Cambodia and Vietnam. This process of 'cutting' or 'flattening' a dendrogram into a set of discrete clusters necessitates specifying either the level at which to cut the dendrogram (in terms of the distribution of distances) or the number of clusters desired. Both of these choices are subjective. In this section, we show how the resulting cluster assignments are sensitive to both the genetic distance and to the algorithm specification (linkage function).

To illustrate the sensitivity of the discrete cluster assignment to the genetic distance, we applied HAC with average linkage (the algorithmic choice used in [9]) to the 1-IBD, 1-IBS and −log₂ IBD distance matrices. Fig 5 tracks how cluster assignment for each isolate varies according to the genetic distance used. The mix of colours present in the stacked barplots of panels B & C of Fig 5 illustrates the sensitivity of cluster assignment to the genetic distance. If there were perfect agreement between cluster assignments, the stacks should each contain one colour only. For each distance matrix, the discrete clusters were constructed using HAC with average linkage. The cluster assignment differs even when the two distances compared are exactly proportional to each other (1-IBD and −log₂ IBD, panels A and C of Fig 5, respectively). We note that cluster assignment using HAC with single linkage is invariant to monotonic transformations of the distance (i.e. HAC with single linkage of a matrix of 1-IBD distances would give an identical clustering arrangement as HAC with single linkage of the matrix of −log₂ IBD distances). However, the single linkage specification is not often used as it produces highly unbalanced tree structures (left or right branching, resulting in unbalanced clusters as shown in panel C of Fig 6). It is important to note that none of the linkage functions, when applied to arbitrary genetic distance matrices have a theoretical basis. The resulting dendrograms have no theoretical guarantees to approximate the unknown phylogenies or pedigrees. They only provide a graphical representation of the genetic distance matrix.

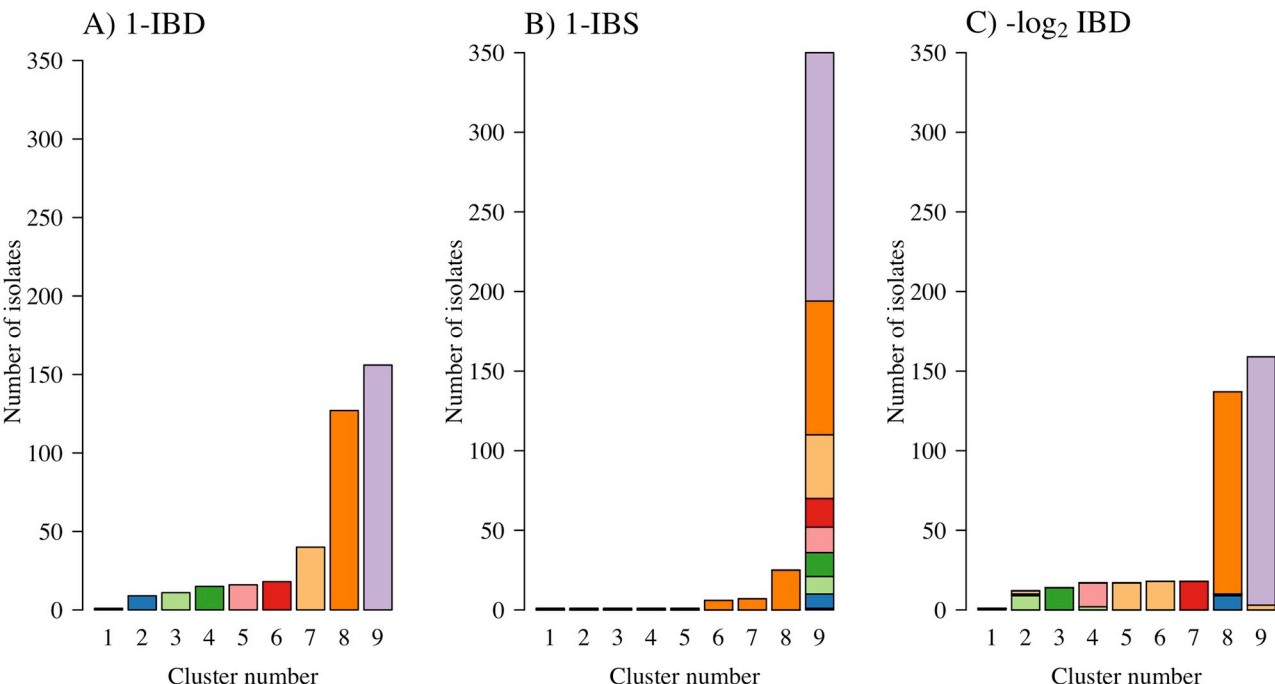

**Fig 5. Tracking the discrete cluster assignments derived from HAC (specification is average linkage) according to the genetic distance matrix used.** Each *P. falciparum* isolate was assigned a colour based on their cluster assignment according to a flattened dendrogram (a dendrogram cut at a given y-axis point) of a clustering arrangement generated by HAC of the 1-IBD distance matrix. In this case the y-axis cut-point was chosen to produce nine distinct clusters (panel A). These colours were then used to track cluster membership when the same HAC algorithm is applied to the 1-IBS and −log₂ IBD distance matrices (panels B and C, respectively).

To illustrate the sensitivity to the linkage function when constructing discrete clusters using HAC algorithms, we computed HAC-based dendrograms using average, complete, and single linkage, and Ward's criterion, applied to the −log₂ IBD distance matrix, which is similar but not identical to the distance matrix used in [9]. We cut each of these dendrograms into nine discrete clusters. Fig 6 tracks how the cluster assignment varies according to the linkage function specified. Fig 6 shows how the clusters discovered under the four algorithmic choices have very few similarities highlighting the strong sensitivity to the algorithm specification.

## Visualising distance matrices using heatmaps ordered with HAC algorithms

HAC algorithms applied to genetic distance matrices are useful for certain tasks. Structure in the genetic distance matrix can be visualised directly via a heatmap. However, the clarity of a heatmap visualisation of the distance matrix is dependent on the ordering of the isolates in the matrix. Finding optimal orderings (known as seriation) whereby all the small distances are concentrated around the diagonal provides improved visualisation. Finding an optimal ordering (e.g. a perfect anti-Robinson matrix [23]) is an NP hard problem (can only be solved in nondeterministic polynomial time) and so heuristic seriation approaches such as HAC are typically used [24]. HAC algorithms are fast and empirically determine a nearly optimal ordering of isolates. As an illustration for how HAC can be used to demonstrate structure within a distance matrix, Fig 7 shows heatmaps for the 1-IBD, 1-IBS, and −log₂ IBD distance matrices, whereby the ordering was determined by HAC with average linkage.

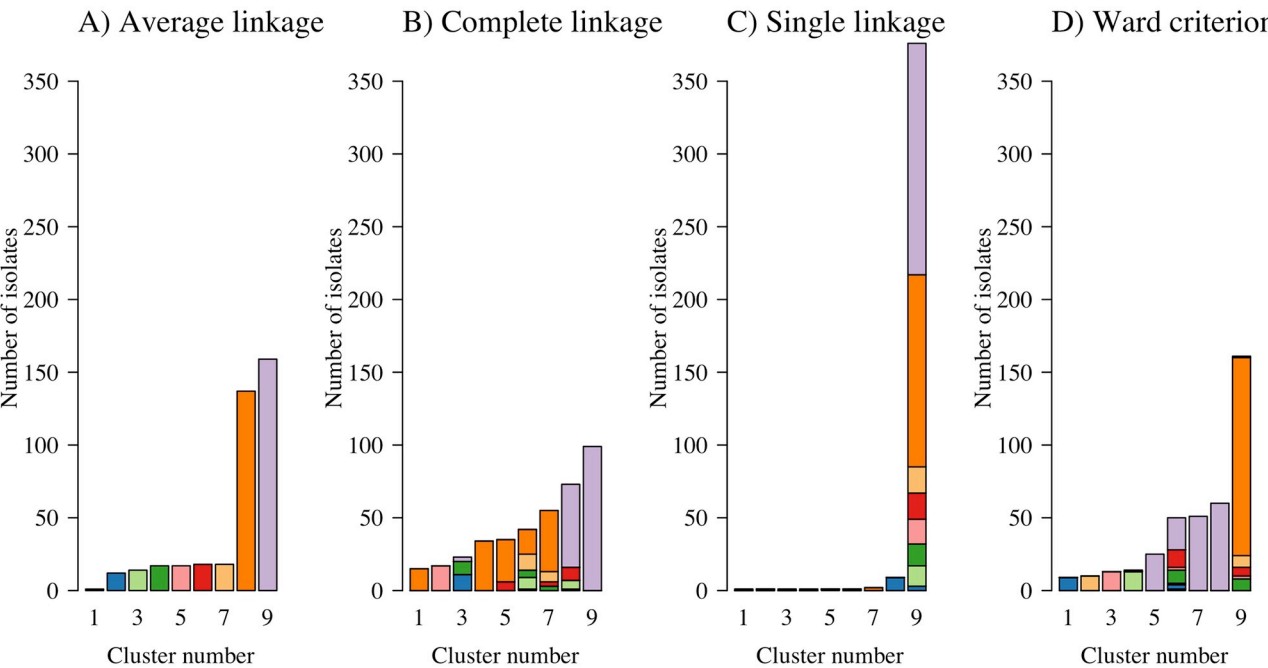

**Fig 6. Tracking the discrete cluster assignment derived from HAC according to the linkage function used applied to the −log₂ IBD distance matrix.** Each *P. falciparum* isolate is assigned a colour based on their cluster assignment from the average linkage algorithm whereby the y-axis cut-point was chosen to produce nine distinct clusters (panel A). Average linkage was arbitrarily chosen as the 'reference' method (any of four linkage functions could be used). These colours are then used to produce stacked barplots for cluster membership derived from three other algorithm specifications (complete, single and Ward's criterion, panels B-D, respectively).

## Reporting recommendations

Box 2 summarises our proposed recommendations around the use of unsupervised machine learning for genetic epidemiology studies in malaria. A key recommendation is that HAC algorithms should not be used to construct discrete clusters as they are sensitive to both the genetic distance and to the algorithmic specification. Analyses that rely on genetic distance matrices

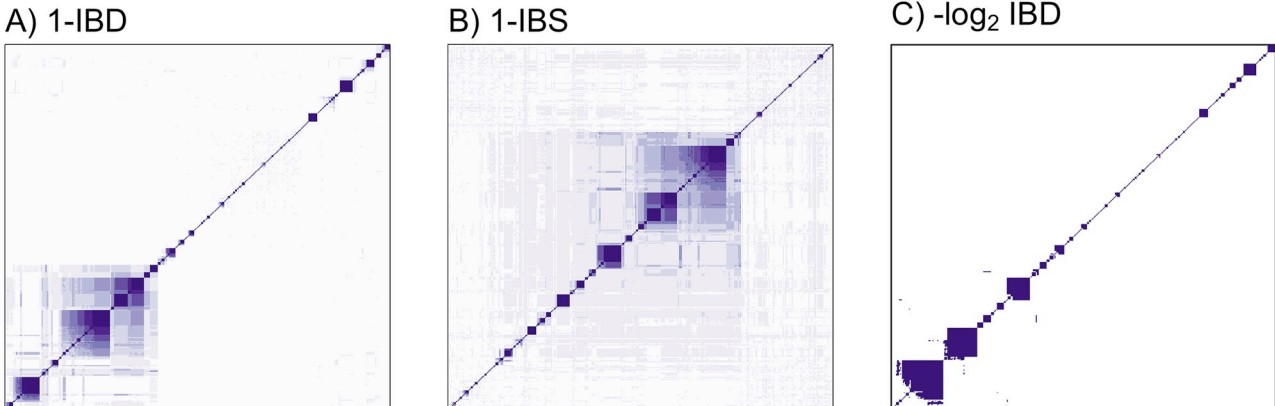

**Fig 7. Heatmaps of genetic distance matrices, whereby isolates have been ordered with the output of the HAC algorithm (average linkage specified here).** The colour shading was chosen by applying nine shades of purple to a uniform grid over the range of observed values (see panels A-C of Fig 2) in the distance matrix. The visual effect of the clustering in the heatmap is sensitive to this specification, for example, a grid over observed quantiles would produce a different visual effect.

## Box 2: Overview of the proposed guidelines for the use of unsupervised learning algorithms in population genetic epidemiology studies of *Plasmodium*.

### Introduction

- *State the overarching scientific question(s) of interest.*

### Methods

- *Justification of unsupervised learning algorithms.* Clearly state why the unsupervised algorithms used were chosen (indeed, the use of any method should be justified) and state precisely the scientific questions they can answer. If the questions that can be answered differ to the overarching question above, we suggest explaining why an inferential method was not used (e.g. because of computational complexity).

- *Hierarchical agglomerative clustering algorithms.* These should not be used to construct discrete population clustering assignments.

- *Sensitivity analysis.* If clustering analyses have been performed, and the presence of biologically meaningful sub-populations are reported, there needs to be a robust analysis examining sensitivity to the genetic distance definition and sensitivity to the underlying methods (both the method used and specification of the method).

- *Publicly available code.* Computational analysis pipelines processing high-dimensional genetic data are usually complex with multiple filtering and data processing steps. The best way to ensure that the analysis is fully reproducible and transparent is to provide readable code. For readability we recommend the use of computational notebooks (e.g. RMarkdown, Jupyter). Although it may not be possible to provide all the raw data used to generate the results, a minimum working example (for example using simulated data or a small subset of isolates) is essential.

- *Publicly available intermediate output/input and meta-data.* Many computational analysis pipelines follow the schematic steps shown in Fig 1. All downstream and sensitivity analyses are thus fully reproducible if the intermediate output/input are provided. Examples of intermediate data summaries include matrices of genetic distances (1-IBS, 1-IBD, $-\log_2$ IBD); IBD segments for all pairs of isolates as output by e.g. *hmmIBD* [25], *isoRelate* [26], *DEploidIBD* [7]; co-ancestry matrix as output by chromosome painting [4]. These intermediate data summaries are privacy-preserving: it is not possible to reverse engineer them and retrieve the genetic data.

- *Publicly available raw data.* Full reproducibility implies access to raw primary data (e.g. non-aligned short reads). Laws and regulations of some countries (e.g. Indonesia) make the availability of raw data impossible. However, intermediate data formats, such as VCF, have been used to ensure reproducibility [27].

### Results

- *Dendrograms.* HAC dendrograms shown as a main result should be accompanied by captions clearly stating that they are not representative of ancestry. We recommend using them primarily as a data visualisation and exploration tool. It should be made

clear (e.g. in Figure captions) that dendrograms do not approximate the underlying phylogeny (or pedigree).

- *Heatmaps*. Genetic distance (similarity) matrices are best represented using heatmaps whereby isolates are ordered using seriation algorithms (this includes HAC algorithms which empirically work well for this purpose).

- *Clustering agreement*. Agreement between clustering algorithms should be represented graphically. We recommend against using tanglegrams for comparing clustering arrangements. Colour coded stacked barplots can help to understand agreement and dis-agreement between discrete cluster membership (e.g. Figs 5 & 6).

## Discussion

- *Speculative reasoning* If the overarching scientific question(s) of interest do not align with the scientific questions answered by the analysis methods, any speculative components to the interpretation should be made clear. This is analogous to causal reasoning based on observational data.

can be made fully reproducible by making the distance matrices available along with the analysis code. In particular we recommend the use of statistical notebooks, such as RMarkdown or Jupyter, which allow for entire analyses to be re-run with little burden to the analyst. Visualisation of the analytic output of clustering methods applied to distance matrices is best done via heatmaps. Dendrograms should not be used as the main analytic output unless they are labelled properly, as their structure is unstable.

We also propose that genetic epidemiology studies that use unsupervised learning algorithms justify explicitly their algorithmic choice(s). Algorithms such as PCoA and HAC are non inferential. The interpretation of their output is restricted. They can be used only to support answers to a given clinical or biological question; they cannot compute the evidence (in a statistical sense) for a particular question. There is therefore a disconnect between the restricted scientific questions that these algorithms can answer and the difficult scientific questions that genetic studies usually pose. It should be made very clear when speculative reasoning is appended to quantitative results from these algorithms to bridge this disconnect. This is analogous to the use of causal reasoning in observational studies: most analyses of observational data have as their ultimate goal providing an answer to a causal question but they are inherently limited to characterising correlation and not causation. For this reason, medical journals usually impose associative language in the results (what the study can answer) and restrict causal reasoning to the discussion of the results (speculation: what the study desires to answer). We propose that genetic epidemiological studies of malaria that feature use of non-inferential algorithms should be subject to analogous restrictions.

## Discussion

The last decade has seen a huge increase in available malaria parasite genetic data, offering profound insights into the genetic epidemiology of both *P. falciparum* and *P. vivax* malaria. However, bespoke analytical tools with which to analyse and interpret these data are still in their infancy. Summarizing dense genetic marker data such as WGS data into a genetic distance

matrix is a key step in many analytical pipelines. The utility of such dimensionality reduction has been demonstrated clearly in human genetics. The method fineSTRUCTURE (operating on the co-ancestry matrix) has provided key insights into recent human evolution, e.g. [28]. In malaria parasite population genetics the choice of genetic distance has been given less consideration than in human population genetics. Recent work has demonstrated the utility of assessing genetic relatedness based on IBD (and its correlate IBS) and using this to measure genetic distance between proximal malaria parasite populations [29, 30]. Output of analyses using an IBD-based genetic distance are interpretable insofar as IBD-based genetic distance explicitly references recent evolutionary events driven by recombination. As for which methods should be applied to parasite genetic distance matrices, our work illustrates that PCoA is a useful exploratory tool. Other alternatives, such as network analytic methods (e.g. [30]) were not explored in this work. We suggest researchers apply PCoA to multiple distance matrices (e.g. 1-IBS, 1-IBD and $-\log_2$ IBD) to characterise sensitivity to the genetic distance. This study highlights why HAC, although useful for ordering samples and visualising distance matrices, should not be used to construct discrete clusters of samples. PCoA and HAC provide a basis from which to explore the key biological questions of interest in the malaria parasite populations under study (e.g. "what is the most likely sequence of events that led to the spread of a single multi-drug resistant *P. falciparum* lineage across Southeast Asia?"), but they do not provide the complete answer.

A possible exception is the program *Relate* which uses a modification of the chromosome painting algorithm and a bespoke HAC algorithm to construct local phylogenies at focal positions along the genome [31]. The output tree structures approximate the unknown local phylogeny under some simplifying assumptions, notably recombination happening strictly at a subset of position (hotspots) and mutations producing the observed derived alleles only occurring once. Heuristic modifications in the HAC algorithm accommodates some departures from these assumptions. However, the utility of the *Relate* program for malaria parasite genetic epidemiology remains to be shown. The scalability of *Relate* is particularly appealing. Understanding how departures from the underlying assumptions can effect the output (the local phylogenetic trees) is essential. The issues with HAC algorithms highlighted in this work should be considered when applying *Relate* to malaria parasite whole genome data.

The emergence and spread of a single multidrug resistant *P. falciparum* lineage across four countries in the eastern Greater Mekong sub-region (GMS) [16, 17] (Box 1) is of serious public health concern. It leaves few options for effective antimalarial therapy in the region. Understanding how drug resistance originates and evolves assists in developing strategies to prevent or curb its spread. The GMS is an area of particular interest for global malaria control and elimination because resistance of *P. falciparum* to all the major antimalarial drugs (chloroquine, sulfadoxine, pyrimethamine, mefloquine, atovaquone, piperaquine and the artemisinins) has originated there and, in the case of chloroquine and sulfadoxine-pyrimethamine, spread across India and Africa with devastating consequences. We believe there are opportunities to improve and extend the use of the available genetic data to answer important questions such as "did artemisinin resistance promote the spread of piperaquine resistance (and if so, by how much)?". Tailor-made statistical methods need to be developed that best utilise the available data to answer these important questions.

Unsupervised machine learning algorithms are powerful tools that can extract structure from vast quantities of complex data and thus generate hypotheses regarding ancestral events. On their own, these methods can neither prove nor disprove an evolutionary model. It is important to note that model-based clustering developed specifically for population genetics— e.g. STRUCTURE—are also easily mis-used. As for PCA, multiple evolutionary models can give rise to identical patterns in the admixture bar plots which are the main analytic output of

these models. The output patterns are also affected by uneven sampling [32]. These bar plots are often mis-interpreted and the strong underlying model assumptions ignored [33]. For the majority of applications, there is no true *K* (the number of clusters inferred by STRUCTURE). It is highly plausible that *Plasmodium* spp. populations strongly diverge from the restrictive model assumptions of the program STRUCTURE. This calls into question its applicability to malaria phylogenetics beyond exploratory data analysis (STRUCTURE is often used in the context of microsatellites, e.g. [34]).

A large class of unsupervised learning algorithms take as input a distance matrix (genetic in our context) and output a summary of the data structure. These algorithms are powerful as they are fast and versatile. However, they have important analytical limitations which may have been under appreciated. In this work we show that two popular methods, PCoA and HAC, are sensitive to the definition of genetic distance and to the algorithmic specification. It is also important to understand the interpretative limitations of the algorithm outputs, which restricts the inferences possible from the data. In some cases the methods used are unable to answer the study questions ([35] gives an example in viral population genetics). It is important that genetic epidemiology studies convey clearly the interpretative limitations of methods applied to the data. The analogy with the use of causal reasoning when analysing observational data is useful. Speculative causality based on associations is an essential scientific exercise, but convention usually restricts this to the discussion of results. At present, in genetic epidemiology studies, speculation often bridges the disconnect between valid interpretation of the analytic output and desired answers to the clinical or epidemiological questions of interest. When this speculation does not make the disconnect clear, it has been described colourfully as 'cargo-cult' science [36, 37], or 'trans-science' [38]; in other words, the use of quantitative methods to give a scientific veneer to speculative results without highlighting the analytical and inferential limitations. Ideally, inferential interpretation would replace speculative interpretation. Statistical models articulated around specific hypotheses allow inferential interpretation but, in many cases, fitting these models is not feasible due to computational limitations (e.g. intractable likelihoods). By using unsupervised clustering algorithms first (to convert a possibly obscure signal in the data into a summary from which a manageable set of hypotheses can be generated), and statistical models second (to do statistical inference under a manageable set of hypotheses), we can offset the advantages and disadvantages of both unsupervised clustering algorithms and statistical models (fast and versatile but non-inferential versus slow and constrained but inferential) and thus address appropriately the questions of interest. The development of statistical models to characterise malaria parasite population ancestry is an open area of research with many exciting methodological challenges (resulting from the complexity of the malaria parasite life cycle). Importantly, if the signal in the data is strong (e.g. as it is in the malaria studies that we use here as running examples), our calls for more sensitivity checks (to address algorithmic sensitivity) and more statistical models (to bridge the inferential disconnect) are not likely to change the overall narrative, but they will make the narrative more robust and, critically, falsifiable (i.e. scientific).

## Materials and methods

### Data

The *P. falciparum* data used in this study are from isolates obtained from patients presenting to rural clinics and health centres in Eastern Thailand (Sri Saket province), Western and North Eastern Cambodia, Southern Lao PDR and Southern Vietnam between 2011 and 2013 during the Tracking Resistance to Artemisinin (TRAC) study [14]. This is a subset of all isolates collected during the TRAC study. It was chosen for simplicity: the parasite isolates come

from a subset of locations that represent a small (and thought to be interconnected in terms of transmission) geographic area of low seasonal malaria transmission. The subset contains WGS data on 52308 single nucleotide polymorphisms (SNPs) that vary across 393 *P. falciparum* isolates that were deemed mononclonal based on a measure of within-host diversity [39] (cutoff at 0.95 of the within-host diversity measure). Only high-quality SNPs from WGS data are routinely used in population genetic analyses due to technical limitations (e.g. see README file https://www.malariagen.net/resource/26). Other polymorphisms, such as indels are useful in highly clonal populations due to their greater mutation rate [40] but currently are not accurately called from short read data.

The data were used to construct matrices of genetic distances between pairs of monoclonal isolates (see next section). Strictly speaking, each 'monoclonal' isolate represents a collection of malaria parasites that can be represented by a single parasite genotype with minimal loss of information. The impact of excluding all infections with complexity above a certain threshold is unclear but has been explored using simulation [13]. In addition, each infection is a subpopulation of parasites with its own ancestry structure. Aggregating this population into a single haplotype (which in fact may not represent any of parasites within the infection) again obscures and possibly biases the estimates of pairwise distance. A better computational pipeline would first phase each infection, and then compute pairwise distances between the $n' \geq n$ haplotypes.

## Matrices of genetic distances between pairs of monoclonal isolates

WGS data are high dimensional. Many computational analysis pipelines first proceed by applying a data processing step to reduce dimensionality. This often takes the form of the construction of a genetic distance or similarity matrix: a *n*-by-*n* matrix that is much smaller than the original *n*-by-*p* data matrix (where *n* is the number of isolates and $p \gg n$ is the number of loci sequenced). Hereafter we refer explicitly to distances only, noting that similarities can be derived from distances and vice versa. A distance or similarity matrix is not necessarily symmetric, e.g. the co-ancestry matrix produced by chromosome painting is in general asymmetric [4] (i.e. the distance is not necessarily a metric in the mathematical sense).

The genetic distances that feature within the distance matrix can be either computed directly as a function of the genetic data (e.g. the fraction of loci where observed alleles differ) or inferred under a statistical model (e.g. the model underpinning chromosome painting [4]). The distance tends to be more interpretable when inferred under a statistical model that references a particular evolutionary process [12], however the inferential process is more involved, and thus possibly less transparent.

A genetic distance can reflect one or more evolutionary processes: mutation, recombination, or both. For example, between a pair of non-recombining organisms (e.g. some viruses, or distinct species), the fraction of loci where observed alleles differ is measured in units of mutation since the most recent common ancestor (MRCA), and can be used to infer the time since the MRCA, e.g. under a model that assumes constant rate mutation. Between a pair of recombining organisms (e.g. malaria parasites), the fraction of loci where observed alleles differ can be measured in units of mutation using data on regions of the genome that do not recombine, e.g. mitochondrial DNA. Otherwise, both mutation and recombination contribute to observed differences. A genetic distance that explicitly references recombination can be obtained via inference under a statistical model.

The best choice of genetic distance depends on which process generates variation over the time-scale most relevant to the scientific question of interest (targeted by the choice of analytical method). For example, among malaria parasites that outcross frequently, distances (either

inferred or not) to which recombination contributes tend to reflect the recent past, as recombination introduces variation at each outcrossing event. Mutation-based distances reflect the more distant past because the region of the genome that does not recombine is small and the mutation rate is relatively low (e.g. compared to that of rapidly evolving viruses). As an aside, population genetic distance measures based on allele frequencies (e.g. $F_{ST}$) can also reflect the more distant past for similar reasons; however, our current focus is on distances that are defined between pairs of isolates, not populations, and measures such as $F_{ST}$ are available only on a population level. The aspect of time-scale is sometimes ignored in genetic analyses of *Plasmodia* but it is essential. S1 Text discusses how pairwise distance measures can be converted into inter-population distances and the implications for the use and limitations of molecular barcodes.

In this study, we use data on 52308 biallelic SNPs from the core *P. falciparum* genome and consider three distinct genetic distances. In the following, $i, j$ index parasite isolates, $k$ indexes the $p$ sequenced loci, and $y_{ik}$ denotes the observed allele for $i$-th isolate at the $k$-th locus.

1. The fraction of loci where observed alleles differ (denoted 1-IBS hereafter):

$$1 - \widehat{\text{IBS}}_{ij} = 1 - \frac{1}{p}\sum_{k=1}^{p} \text{IBS}_{ijk}, \tag{1}$$

where $\text{IBS}_{ijk} = 1$ if $y_{ik} = y_{jk}$ and 0 otherwise. In other words, $\text{IBS}_{ijk} = 1$ if observed alleles $y_{ik}$ and $y_{jk}$ are identical-by-state (IBS). Each $\text{IBS}_{ijk}$ is observed. As such, $1 - \widehat{\text{IBS}}_{ij}$ can be computed directly as a function of genetic data. Note that Eq (1) does not feature a linkage disequilbrium correction (used elsewhere e.g. [41]). It does not feature allele frequencies and it does not reference any particular process. As stated above, both mutation and recombination contribute to its value. Although it is not inferred, 1-IBS is thus a distance to which recombination contributes. Its expectation is a linear function of relatedness [12].

2. Unrelatedness estimate (denoted 1-IBD):

$$1 - \hat{r}_{ij} = 1 - \mathbb{P}(\text{IBD}_{ijk} = 1) \text{ for all } k = 1, \dots, p, \tag{2}$$

where $\text{IBD}_{ijk} = 1$ if $y_{ik} = y_{jk}$ and are both descended from a recent common ancestor, 0 otherwise. In other words, $\text{IBD}_{ijk} = 1$ if observed alleles $y_{ik}$ and $y_{jk}$ are identical-by-descent (IBD). Each $\text{IBD}_{ijk}$ is unobserved. As such, each relatedness estimate, $\hat{r}_{ij}$, is inferred under a statistical model. The statistical model is typically a hidden Markov model (HMM) that features allele frequencies and explicitly references sexual recombination to some extent, e.g. [25]. We used R code available via [12] to estimate relatedness. Unrelatedness is thus an inferred genetic distance to which recombination contributes.

3. An approximation of the number of outcrossed generations since the MRCA (denoted $-\log_2\text{IBD}$):

$$-\log_2 \hat{r}_{ij}. \tag{3}$$

This approximation is based on the expectation that relatedness is halved each time fully outbred parasites recombine sexually. The number of outcrossed generations is less than or equal to the number of life cycle generations: although the malaria parasite life cycle involves an obligate stage of sexual recombination, malaria parasites can self. When there is outcrossing at each life-cycle, this distance measure is a linear function of the number of life-cycles.

As a sensitivity analysis, we compared the PCoA of these three genetic distance matrices with a PCoA of the co-ancestry matrix output from the program fineSTRUCTURE [4]. The co-ancestry matrix is a similarity matrix (increasing values show increasing similarity between pairs). Each entry $x_{ij}$ of the co-ancestry matrix is proportional to the number of 'chunks' in the genome donated from isolate $j$ to isolate $i$. fineSTRUCTURE estimates the co-ancestry matrix with the chromosome painting algorithm. This uses a hidden Markov model to paint each isolate as a piece-wise combination of all other $n - 1$ isolates. A chunk is defined as a contiguous region of the genome unbroken by recombination between $i$ and $j$.

## Unsupervised machine learning algorithms in statistical genetics

Unsupervised learning encompasses many different algorithms and model families, most of which are considered to be machine learning methods but originally developed in statistics. In this work we consider two main algorithmic classes, PCoA and HAC. PCoA is the linear version of the more general approach to dimensionality reduction, known as multidimensional scaling. Some analyses of malaria parasite population genetics have used non-linear multidimensional scaling (e.g. [42] uses t-distributed stochastic neighbour embedding [43]) but the advantages of non-linear methods over the linear counterparts are unclear, therefore we consider only PCoA. Supervised learning methods have also been suggested for population genetics [44] but these have been rarely applied to parasite population genetics and are outside the scope of this work. PCoA and HAC are both non-inferential (i.e. they do not have parameters inferred from data via a likelihood function), but we note that some unsupervised learning methods are inferential, e.g. STRUCTURE.

**Principal coordinates analysis.** Principal components analysis (PCA), also referred to as eigenanalysis, is an essential method for the assessment of population structure [2]. PCA usually refers to an eigenanalysis of the sequence covariance matrix. However, it is possible to do an eigenanalysis on any genetic distance matrix, for any user-defined genetic distance. This is referred to as principal coordinates analysis (PCoA).

PCA on the covariance matrix has a solid theoretical footing in statistical genetics [11, 45]. Different patterns of divergent evolution and admixture lead to specific patterns in the top principal components. A genealogical interpretation can be given to the output of a PCA under the coalescent model, but with limitations [11]. Multiple histories may lead to the same PCA structure and uneven sampling can skew the output severely [11, 46]. There does not exist currently the same theoretical footing for PCoA on arbitrary genetic distances matrices. Therefore PCoA is primarily a technique for visualising structure in the genetic distance matrix and for reducing dimensionality of the distance matrix to a few eigenvectors with the largest eigenvalues (proportion of variance explained) that approximates the number of dimensions on which the data vary. The detection of clusters from PCoA or PCA is sometimes estimated directly from the first two (or more) principal components of the distance matrix, as reported recently in a large analysis of *P. falciparum* parasites from the African continent [42].

**Hierarchical agglomerative clustering.** Unsupervised clustering methods are important tools in machine learning and statistics more broadly. The goal of an unsupervised clustering algorithm is to partition a set of observations (agnostic of the data type, whether genetic or other) into a discrete number of subsets (clusters). This partition should in theory minimise the distances between members of the same clusters (intra-cluster), and maximise the distances between members of different clusters (inter-cluster). The main class of unsupervised clustering algorithms used in malaria genetic epidemiology is hierarchical agglomerative clustering (HAC). This is a class of bottom-up clustering algorithms which operate directly on distance matrices. HAC includes as special cases UPGMA (UPGMA stands for unweighted pair

group method with arithmetic mean), and neighbour-joining. By bottom-up, we mean that at the first iteration of the algorithm, each observation is in its own cluster. At subsequent iterations, the two 'nearest' distinct clusters are merged. At the first iteration of the algorithm, 'nearest' is defined by the distance matrix, subsequently it is defined by the linkage function applied to the distance matrix (see below). Therefore, a HAC algorithm necessitates specification of the following:

- A definition of distance between pairs of observations,

- A linkage function that determines the distance between sets of observations (the inter-cluster distance), which is used to assess a proposed joining of clusters.

HAC algorithms are 'greedy' (greedy in computer science refers to the process of local optimisation, i.e. only looking one step ahead at each iteration of the algorithm). The linkage function determines the update at each iteration and is key to the specification of the algorithm. The most common choices reported in the literature are to use average linkage (average inter-cluster distance), single linkage (the minimum inter-cluster distance, known as closest neighbours), complete linkage (the maximum inter-cluster distance, known as furthest neighbours), or Ward's criterion (the total within cluster variance, applicable to cases where the distance matrix is given by squared Euclidean distances). HAC algorithms do not output a set of discrete clusters explicitly but instead they provide a set of nested clustering arrangements. This nested clustering arrangement can be visualised as a dendrogram which depicts *a possible* underlying data structure based on the distance matrix. The y-axis scale on the dendrogram (corresponding to the branch lengths) is proportional to the inter-cluster distance. It is possible to 'flatten' this dendrogram by choosing a cut-off point on the y-axis. This cut is either done by specifying a number of desired clusters and thus finding an appropriate value on the y-axis to obtain this number, or by just specifying a particular value on the y-axis (in units of inter-cluster distance). Importantly, both are arbitrary choices. We note that recent work provides a method that can test for a dependence between the tree structure and a given leaf variable (e.g. isolate phenotype coded as a binary or continuous variable) [22]. This explicitly takes into account all possible clustering structures and avoids having to arbitrarily choose a cut-off threshold.

We note that in general it is not possible to give an evolutionary interpretation to the output of HAC for a particular choice of distance metric and linkage function. We know of one exception, however: the program *Relate* [31]. *Relate* uses a modified version of the Li and Stephens hidden Markov model such that for a given position on the genome, each entry $(i, j)$ of the output distance matrix corresponds to the number of derived alleles present in sample $i$ and not sample $j$ (by derived allele we mean a mutation that happened on the branch between $i$ and the MRCA of $i, j$, this necessitates specifying which allele is ancestral and which is derived). The bespoke HAC algorithm used by *Relate* makes use of this interpretation in order to construct a binary coalescence tree, guaranteed to be correct if the assumptions of the model are met. The influence of the simplifying assumptions in the model (recombination occurring only at hotspots, and mutations only occurring once) is likely to vary across organisms.

Neighbour-joining is a special case of a HAC algorithm [47]. This was designed specifically for the construction of non-rooted trees based on an evolutionary distance measured in units of mutation. The motivation for the neighbour-joining algorithm is finding the most parsimonious tree with respect to the data, whereby parsimonious means that the fewest number of evolutionary events (e.g. mutation events) are needed to explain the differences between observations. We note that even in absence of recombination, neighbour-joining trees cannot be interpreted as phylogenetic trees. Construction of a phylogenetic tree necessitates the

specification of the direction of change (ancestral versus derived alleles), thus a phylogenetic tree is by definition rooted [48]. The neighbour-joining algorithm starts from a star-like tree structure and then iteratively joins 'neighbours' where neighbours are determined by a joining event that minimises the total length of all the tree branches in the subsequent tree (the minimum total branch length here implies the most parsimonious tree representation). When attempting to determine ancestry the objective is to build a phylogenetic structure. But in absence of rooting (usually done by incorporating an out-group) and where differences are mainly driven by recombination and not mutation or drift (e.g. sexually recombining malaria parasites), genetic distance based on IBD (or its correlate, IBS) is not linearly proportional to evolutionary genetic distance driven by mutation.

## Statistical analysis

All statistical analyses were performed in R version 3.6.0 [49]. PCoA and HAC were applied to three distance matrices each featuring different distances outlined above: $1-$IBS (Eq (1)), $1-$IBD (Eq (2)), and $-\log_2$ IBD (Eq (3)). For the $-\log_2$ IBD genetic distance, if relatedness was estimated to be exactly zero (i.e. 1-IBD = 1), then we replaced it by the approximated lower limit of quantification of IBD (the lowest observed non-zero IBD value). The PCoA was performed using the R function *cmdscale*. HAC was performed using the R package *fastcluster* [50]. Four separate specifications of the HAC algorithm were considered, each using a different linkage function: average linkage, complete linkage, single linkage, and Ward's criterion. The function *cutree* (R package *dendextend* [51]) was used to collapse the resulting dendrograms into nine discrete clusters. The choice of nine was arbitrary: there is no principled approach for choosing the number of clusters into which to cut the dendrograms. The co-ancestry matrix was computed using fineSTRUCTURE version 4.1.0. This was computed using an effective population size parameter equal to 1000 (argument -n in fineSTRUCTURE).

The three genetic distance matrices (data summaries) and code used to generate the results reported here are openly available at the following github repository: https://github.com/jwatowatson/sensitive-plasmodium-structure.

## Supporting information

**S1 Text. A note on the use of IBD for inter-population comparisons and the construction of molecular barcodes.**
(PDF)

## Acknowledgments

We thank Dominic Kwiatkowski for suggesting improvements to the manuscript and for pointing out the relevance of the *Relate* program. We thank Sungsik Kong and Santiago J. Sánchez-Pacheco for helpful comments on the first preprint version of the manuscript.

## Author Contributions

**Conceptualization:** James A. Watson, Aimee R. Taylor, Chris C. Holmes.

**Data curation:** James A. Watson.

**Formal analysis:** James A. Watson.

**Funding acquisition:** Nicholas J. White.

**Investigation:** James A. Watson.

**Methodology:** James A. Watson, Aimee R. Taylor.

**Resources:** Elizabeth A. Ashley, Arjen Dondorp, Nicholas J. White.

**Software:** James A. Watson.

**Supervision:** Aimee R. Taylor, Caroline O. Buckee, Nicholas J. White, Chris C. Holmes.

**Validation:** James A. Watson, Aimee R. Taylor, Elizabeth A. Ashley, Arjen Dondorp, Caroline O. Buckee, Nicholas J. White, Chris C. Holmes.

**Visualization:** Aimee R. Taylor.

**Writing – original draft:** James A. Watson, Aimee R. Taylor.

**Writing – review & editing:** James A. Watson, Aimee R. Taylor, Elizabeth A. Ashley, Arjen Dondorp, Caroline O. Buckee, Nicholas J. White, Chris C. Holmes.

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
