## [Decision Letter · Decision Letter 0]

30 May 2020

Dear Dr Watson,

Thank you very much for submitting your Research Article entitled 'A cautionary note on the use of machine learning algorithms to characterise malaria parasite population structure from genetic distance matrices' to PLOS Genetics. Your manuscript was fully evaluated at the editorial level and by independent peer reviewers. The reviewers appreciated the attention to an important problem, but raised some substantial concerns about the current manuscript. Based on the reviews, we will not be able to accept this version of the manuscript, but we would be willing to review again a much-revised version. We cannot, of course, promise publication at that time.

If you decide to revise the manuscript for further consideration at PLOS Genetics, please aim to resubmit within the next 60 days, unless it will take extra time to address the concerns of the reviewers, in which case we would appreciate an expected resubmission date by email to plosgenetics@plos.org.

[LINK]

We are sorry that we cannot be more positive about your manuscript at this stage. Please do not hesitate to contact us if you have any concerns or questions.

Yours sincerely,

Giorgio Sirugo

Associate Editor

PLOS Genetics

Scott Williams

Section Editor: Natural Variation

PLOS Genetics

Reviewer's Responses to Questions

**Comments to the Authors:**

Reviewer #1: review uploaded

Reviewer #2: I am aware of researchers who have spent alot of time trying to (unsuccessfully) reproduce Plasmodium falciparum and P. vivax sequence-based population structure analyses and conclusions, including some of those presented within the publications cited in the manuscript. The underlying difficulties in some published work seem to be multi-factorial, including methodologies applied being opaque or non-robust, data being selectively removed, collapsed or not made available (e.g. Indonesian sequence data) and, as per the manuscript, some over-interpretation of results and unawareness of assumptions. Practices are changing and methodologies improving, including with some uptake in the use of IBD approaches, and therefore the proposed manuscript is important for future robust population structure analysis; especially as such analyses could be guiding disease control activities. To strengthen the manuscript, the best practice guidelines therein, and potential impact, I have suggested some edits:

(1) There is a need for additional references in the context of malaria biology or epidemiology. For example, statements like “..unlike humans, malaria parasites can self (recombination between genetically identical male and female gametes), and the rate of selfing varies with transmission intensity.” (line 49) could be supported by a citation. In addition, there is a need for a few sentences discussing different types of machine learning (ML) methods. It seems the manuscript has narrow scope in the ML approaches applied, so some discussion of other approaches that can be used (e.g. random forests, support vector machines, etc. for classification) and examples will be useful for the reader. Perhaps, these approaches be more explicit in Figure 1.

(2) One of the issues with analyses of Plasmodium sequencing direct from clinical samples is the multiplicity of infection and potential co-infections (e.g. P. falciparum and malariae co-infections). For the former, the Fws and estMOI software have been applied to triage out samples with >1 clones. It may be worth in the discussion or methods sections (where Fws is mentioned) stating in greater detail how MOI and co-infections can affect genetic clustering and IBD analyses.

(3) The starting point for the analyses in the manuscript is a set of SNPs and samples, which have gone through a bioinformatic pipeline. In general, it is extremely difficult to generate exactly the same dataset even with a detailed Methods section. Hence, your call for greater availability of intermediate datasets is a good one. Equally, all underlying raw data needs to be available, and its unavailability is a problem in some P. falciparum and P. vivax genomics studies, where some populations (e.g. Indonesia) appear in print, but their raw data is unavailable. Whilst, I do not want the authors to discuss politics, there is a need for a general sentence about the need for complete data availability and citing some examples of P. falciparum and P. vivax studies where only partial raw sequence datasets were made available.

(4) Related to (3), there is a clear need for greater transparency in bioinformatic and statistical/analytical pipelines (this could be included in Panel 1). The effects of allele frequency thresholding, SNP detection algorithms, and how missing genotype values are handled, could also impact on population genetic and structure analyses. It would be worth discussing this briefly.

(5) As stated, there are at least 3 software tools available for IBD analyses. Whilst the focus of the manuscript is not on inter-population comparisons (Fst analyses are cited for this), there is a need to discuss whether and how IBD could be used for such analysis. It is a natural extension from your work to identify the genetic regions that are driving population structure, and by providing a brief discussion of this would lead to greater impact, especially to inform those working on molecular barcodes for geographical and transmission classification. More generally, transmission applications could be referenced.

(6) The focus of the manuscript is mainly on SNP analysis, but there are also indel variants, and Figure 1 also mentions microsatellites. Should we include indels to improve population structure resolution and inference? How could indels be included into the ML approaches? Also, a quick pubmed search revealed many papers performing population STRUCTURE analyses using microsatellites where there is an over-interpretation of “K” (as highlighted, lines 476-). Given the large number of studies, it would make sense to include a very recent example to highlight your point and demonstrate that it is a contemporary issue (the most recent one I found from a pubmed search was PMID: 32379762, but feel free to identify an alternative appropriate example).

(7) The dataset used for the analysis is quite small (N=393; covering years 2011-13), especially compared to the much larger datasets currently available. Whilst, I am not suggesting a revised analysis with the Pf3K data from the same locations post-2013, it would be useful to discuss the impact and use of additional data (e.g. for machine "learning"), and any new insights in light of subsequent new control measures.

(8) Please check that gene names are italicised.

Reviewer #3: This is an interesting and thoughtful paper highlighting the lack of critical thinking around the application of ML algorithms in analyses of malaria parasite genetic data.

Comments:

I struggled to get a sense of the questions that the authors had in mind in general. In the introduction and discussion there are isolated examples of the questions addressed by the methods. For example, p1 “an important goal in malaria parasite genetic epidemiology is inference of the full ancestral recombination graph” p2 “Many questions of clinical and public health relevance, for example, interpreting reduced haplotype diversity as a selective sweep”, p2 “characterizing population structure”, p13 “construct discrete clusters of samples” and p13 “what is the most likely sequence of events that led to the spread of a single multi-drug resistant P falciparum lineage across Southeast Asia”. There seems to be a swing from hugely broad to very specific questions – I can see why such a problem exists since there are many possible questions. It might be helpful to give some boundaries to the scope of the questions, or to categorize the types of questions, describe the categories and give specific examples within a category.

There is a long paragraph in the introduction on the epidemic of multi-drug resistant parasites in the GMS. This was interesting, but I felt I had wandered into another paper. It is told from the point of view of someone interested in drug resistance, but for this paper it might be more relevant to focus on the different methods and the different conclusions reached, or to introduce it as a motivating example in a Panel.

The rationale for the choice of example is not explained. It is interesting, has available data and gives different results by method - but perhaps it is a particular case. Since the paper is talking in general terms for a wider scope of questions, it may be interesting to consider if there is a set of conditions under which the results for the three methods would be the same.

Chromosome painting is mentioned several times. It would be helpful to briefly describe how this works.

There did not seem to be any mention of statistical models to infer networks of infection. Are these outside the scope of the questions considered?

p5 Genetic distance measure 3. It would be helpful to mention how this measure behaves when there is recombination.

NP - is this written out somewhere? (p12)

**Have all data underlying the figures and results presented in the manuscript been provided?**

Reviewer #1: Yes

Reviewer #2: Yes

Reviewer #3: Yes

PLOS authors have the option to publish the peer review history of their article (what does this mean?). If published, this will include your full peer review and any attached files.

Reviewer #1: No

Reviewer #2: No

Reviewer #3: No

---

## [Decision Letter · Decision Letter 1]

8 Aug 2020

Dear Dr Watson,

We are pleased to inform you that your manuscript entitled "A cautionary note on the use of unsupervised machine learning algorithms to characterise malaria parasite population structure from genetic distance matrices" has been editorially accepted for publication in PLOS Genetics. Congratulations!

Yours sincerely,

Giorgio Sirugo

Associate Editor

PLOS Genetics

Scott Williams

Section Editor: Natural Variation

PLOS Genetics

Comments from the reviewers (if applicable):

Reviewer's Responses to Questions

**Comments to the Authors:**

Reviewer #1: The authors have satifactorily duel on most of the issues raised. Use of panels for further detailing has improved the paper.

Reviewer #3: The authors have revised the manuscript well and I have no further comments.

Minor comment:

The sentence starting L90 is awkwardly worded.

**Have all data underlying the figures and results presented in the manuscript been provided?**

Reviewer #1: Yes

Reviewer #3: Yes

PLOS authors have the option to publish the peer review history of their article (what does this mean?). If published, this will include your full peer review and any attached files.

Reviewer #1: **Yes: **Alfred Amambua-Ngwa

Reviewer #3: No

**Data Deposition**

http://datadryad.org/submit?journalID=pgenetics&manu=PGENETICS-D-20-00579R1

**Press Queries**

---

## [Editor Report · Acceptance letter]

1 Oct 2020

PGENETICS-D-20-00579R1 

A cautionary note on the use of unsupervised machine learning algorithms to characterise malaria parasite population structure from genetic distance matrices 

Dear Dr Watson, 

We are pleased to inform you that your manuscript entitled "A cautionary note on the use of unsupervised machine learning algorithms to characterise malaria parasite population structure from genetic distance matrices" has been formally accepted for publication in PLOS Genetics! Your manuscript is now with our production department and you will be notified of the publication date in due course.

With kind regards,

Matt Lyles

PLOS Genetics

On behalf of:
